# Berberine inhibits free fatty acid and LPS-induced inflammation *via* modulating ER stress response in macrophages and hepatocytes

**Yanyan Wang**[1,2], **Xiqiao Zhou**[2,3], **Derrick Zhao**[2], **Xuan Wang**[2], **Emily C. Gurley**[2], **Runping Liu**[2], **Xiaojiaoyang Li**[2], **Phillip B. Hylemon**[2], **Weidong Chen**[1]*, **Huiping Zhou**[2]*

1 School of Pharmaceutical Science, Anhui University of Chinese Medicine, Hefei, China, 2 Department of Microbiology and Immunology and McGuire Veterans Affairs Medical Center, Virginia Commonwealth University, Richmond, Virginia, United States of America, 3 Department of Gastroenterology, The First Affiliated Hospital of Nanjing Medical University, Nanjing, Jiangsu, China

* anzhongdong@126.com (WC); huiping.zhou@vcuhealth.org (HZ)

## Abstract

Inflammation plays an essential role in the pathogenesis of non-alcoholic fatty liver disease (NAFLD). Berberine (BBR), an isoquinoline alkaloid isolated from Chinese medicinal herbs, has been widely used to treat various diseases, including liver diseases for hundreds of years. The previous studies have shown that BBR inhibits high fat-diet-induced steatosis and inflammation in rodent models of NAFLD. However, the underlying molecular mechanisms remain unclear. This study is aimed to identify the potential mechanisms by which BBR inhibits free fatty acid (FFA) and LPS-induced inflammatory response in mouse macrophages and hepatocytes. Mouse RAW264.7 macrophages and primary mouse hepatocytes were treated with palmitic acid (PA) or LPS or both with or without BBR (0–10 μM) for different periods (0–24 h). The mRNA and protein levels of proinflammatory cytokines (TNF-α, IL-6, IL-1β, MCP-1) and ER stress genes (CHOP, ATF4, XBP-1) were detected by real-time RT-PCR, Western blot and ELISA, respectively. The results indicated that BBR significantly inhibited PA and LPS-induced activation of ER stress and expression of proinflammatory cytokines in macrophages and hepatocytes. PA/LPS-mediated activation of ERK1/2 was inhibited by BBR in a dose-dependent manner. In summary, BBR inhibits PA/LPS-induced inflammatory responses through modulating ER stress-mediated ERK1/2 activation in macrophages and hepatocytes.

## 1. Introduction

Non-alcoholic fatty liver disease (NAFLD) is one of the most common liver diseases. An increasing body of evidence suggests that the disease progression of NAFLD is closely associated with inflammation, obesity, insulin resistance, and metabolic syndrome [1,2]. However, the exact molecular/cellular mechanisms underlying NAFLD remain obscure and the effective therapeutic strategies are still limited.

**Data Availability Statement:** All the data are included in paper.

**Funding:** This study was supported by National Institutes of Health Grant R01 DK104893 and

R01DK-057543 (HZ and PBH); VA Merit Award I01BX004033 (HZ) and 1I01BX001390 (HZ); Research Career Scientist Award (HZ, IK6BX004477); Massey Cancer Center research fund (HZ); Jiangsu Province Special Program for Young Medical Talent Grant (XZ, QNRC2016568); National Natural Science Foundation of China (XZ,81570522).

**Competing interests:** The authors declare no competing financial interest.

**Abbreviations:** BBR, berberine; ELISA, enzyme-linked immunosorbent assay; ER, endoplasmic reticulum; FFA, free fatty acid; LPS, lipopolysaccharide; NAFLD, nonalcoholic fatty liver disease; PA, palmitic acid; PERK, protein kinase RNA-like endoplasmic reticulum kinase; RBP, RNA-binding protein; SFA, saturated fatty acid; UPR, unfolded protein response; WT, wild type.

The elevated circulating levels of lipopolysaccharide (LPS) due to disruption of intestinal barrier function and free long-chain fatty acids (FFA) are implicated in insulin resistance and systemic inflammation, both of which are positively associated with the development and progression of NAFLD [3,4]. Although the mechanism by which LPS/FFA induce hepatic lipotoxicity is still not fully understood, LPS/FFA-induced expression of inflammatory cytokines, such as TNF-α and IL-6, and activation of the endoplasmic reticulum (ER) stress signaling pathway, known as the unfolded protein response (UPR), are major contributors [5–8].

Berberine (BBR), an isoquinoline alkaloid isolated from many medicinal herbs, is one of the widely used traditional Chinese medicines and has been used to treat various infectious disorders for more than 3,000 years [9]. During the last few decades, many studies have shown that BBR exerts various beneficial effects on cardiovascular and metabolic diseases [10]. It also has been reported that BBR can prevent NAFLD disease progression by regulating multiple metabolic pathways and reducing inflammation response [11]. We have previously reported that BBR could inhibit HIV protease inhibitor-induced ER stress and TNF-α and IL-6 expression through regulating the RNA-binding protein (RBP) HuR in macrophages [12].

In this study, we specifically examined the effect of BBR on PA/LPS-induced inflammatory response in macrophages and hepatocytes. The results indicated that BBR significantly inhibited PA/LPS-induced inflammatory response *via* modulating ER stress and ERK1/2 activation.

## 2. Materials and methods

### 2.1. Materials

Antibodies against phospho-ERK1/2, ERK1, ERK2, CHOP, ATF-4, XBP-1, IL-1β, ATF6, GRP78, IRE1α, β-Actin were from Santa Cruz Biotechnology (Santa Cruz, CA, USA). The detailed information of the antibodies was listed in S1 Table. Berberine (BBR), lipopolysaccharides (LPS), and Palmitic acid (PA) were purchased from Sigma (St. Louis, MO, USA). Bovine Serum Albumin Fraction V, heat shock, fatty acid-free, was from Roche (Roche Diagnostics GmbH, Mannheim, Germany). All cell culture media were purchased from Thermo Fisher (Waltham, MA, USA).

### 2.2. Cell culture and treatment

Mouse 264.7 macrophages (ATCC, Rockville MD, USA) were cultured as previously described [13]. BBR was dissolved in DMSO while LPS was dissolved in the culture medium. PA was firstly dissolved in ethanol at 200 mM followed by combination with 10% FFA-free, low-endotoxin BSA, giving a final concentration of 5 mM. The working solution was prepared fresh by diluting the stock solution (1:10) in the culture medium.

### 2.3. Isolation of primary mouse hepatocytes

Primary mouse hepatocytes were isolated from C57BL/6 wild type mice (male, 6–8 weeks old, from Jackson Laboratories, Bar Harbor, ME, USA) by the collagenase-perfusion technique, which has been previously described [14]. Mice were anesthetized with continuous 2% isoflurane in $O_2$ (500 cc/min) during the isolation. After that, mice were euthanized by cervical dislocation. The procedures for isolation of primary mouse hepatocytes were approved by the Virginia Commonwealth University Institutional Animal Care and Use Committee (Approved protocol number: AD1001773). Hepatocytes were plated at the collagen-coated 60-mm dish or 6-well plate in serum-free Williams E medium containing penicillin, dexamethasone (0.1μM), and thyroxine (1μM) [15].

## 2.4. RNA isolation and real-time Quantitative RT-PCR

Total RNA was isolated using TRIzol Reagent (QIAGEN, Valencia, CA, USA) following the manufacturer's protocol. The first-strand cDNA was reverse transcribed and Quantitative PCR analysis of relative mRNA levels of target genes was performed, as previously described [16]. The mRNA levels of CHOP, ATF4, XBP-1s, XBP-1us, TNF-α, IL-6, IL-1β, and MCP-1 were quantified by real-time PCR using gene-specific primers. Primer sequences used are provided in S2 Table.

## 2.5. Enzyme-linked immunosorbent assay (ELISA) of TNF-α, IL-6, MCP-1

Mouse RAW264.7 macrophages were pre-treated with BBR (5 μM) for 1 h, then treated with PA (0.25 mM) or LPS (25 ng/mL) or PA plus LPS for 6 h. Wild type-derived primary hepatocytes were pre-treated with BBR (5 μM) for 1 h, then treated with PA (0.25 mM) or LPS (25 ng/mL) for 6h. At the end of the treatment, the culture medium was collected and centrifuged to remove the cell debris. The protein levels of TNF-α, IL-6, and MCP-1 in the media were measured using mouse TNF-α, IL-6, and MCP-1 ELISA Max™ Set Deluxe Kits (Biolegend, San Diego, CA, USA) as previously described [17,18]. The total protein concentrations of the viable cells were measured using Bio-Rad Protein Assay reagent and Bradford protein assay. Total amounts of the TNF-α, IL-6, and MCP-1 were normalized to the total protein amount of the viable cells and expressed as pg/mg protein.

## 2.6. Western blot analysis

Total cellular proteins were prepared using cold RIPA buffer as previously described [19]. Protein concentration was measured using the Bio-Rad Protein Assay reagent. Proteins were resolved by 10% SDS-PAGE and transferred to nitrocellulose membranes. After blocking with 5% nonfat milk in TBS-T, the membranes were incubated with the primary antibodies overnight at 4℃ followed by detection using horseradish peroxidase-conjugated secondary antibody. The antibody-antigen complexes were detected using the ECL system (Thermo Scientific, Rockford, IL, USA). The density of immunoblotted bands was analyzed using Bio-Rad Image Lab computer software and normalized with β-Actin [15].

## 2.7. Statistical analysis

Results are expressed as the mean ± SEM and are from at least three independent experiments. One-way analysis of variance was performed to compare the differences between multiple groups using GraphPad Prism 5.0 (GraphPad Software Inc., San Diego, CA, USA). A value of $p \leq 0.05$ was considered statistically significant.

## 3. Results

### 3.1. Effect of BBR on PA and LPS-induced upregulation of the proinflammatory cytokines in RAW264.7 cells

Activation of the inflammatory response is a critical driving force of NAFLD disease progression. Macrophages are the major sources of proinflammatory cytokines and chemokines, such as TNF-α, IL-6, IL-1β, and MCP-1 [20]. Our previous studies reported that BBR could inhibit HIV protease inhibitor-induced inflammatory response *via* modulating ER stress in mouse J774A.1 macrophages [12]. To investigate the potential anti-inflammatory properties of BBR on PA and LPS-induced inflammation in macrophages, we first determined the mRNA expression levels of the proinflammatory cytokines and chemokine, including TNF-α, IL-6,

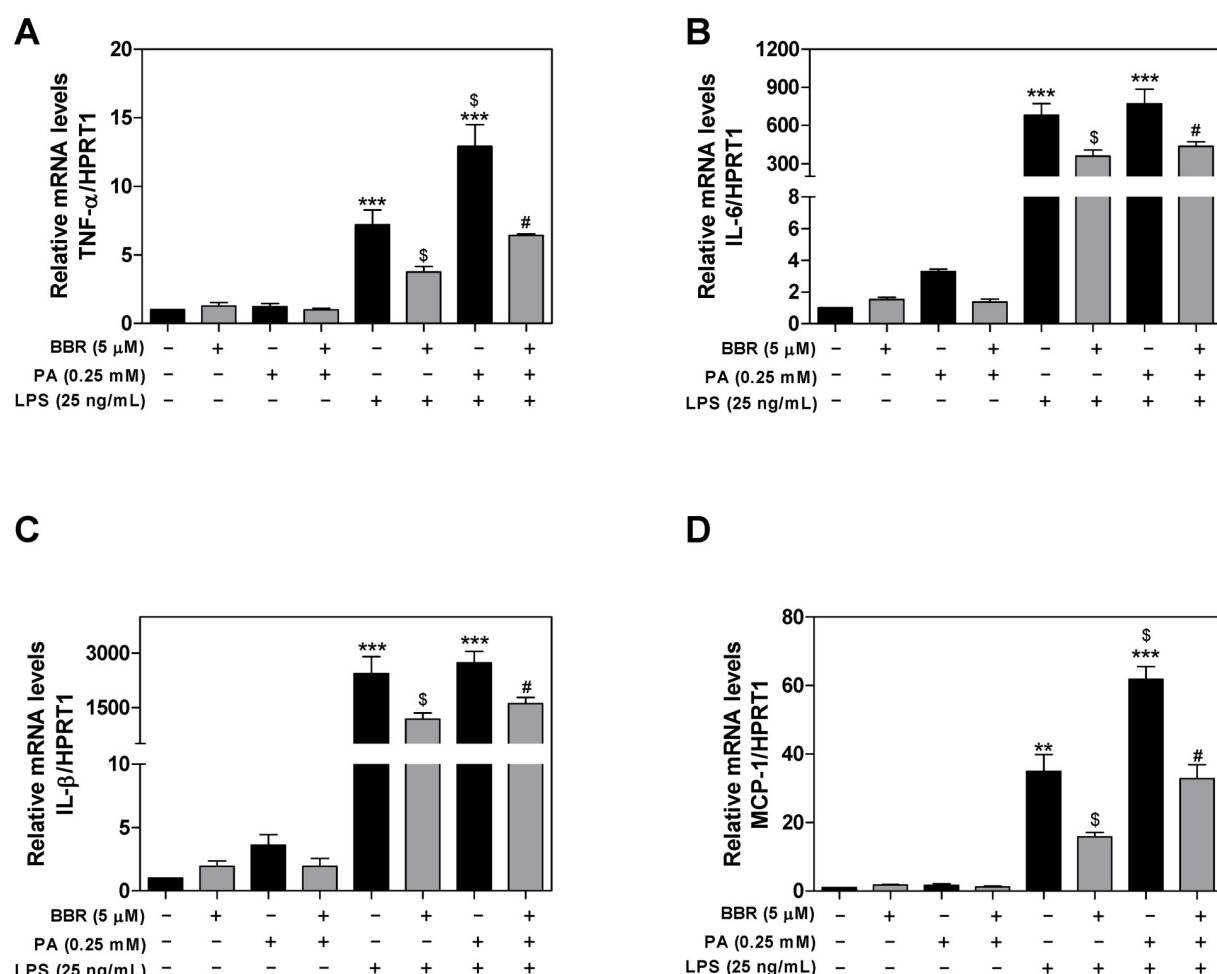

**Fig 1. Effect of BBR on PA and LPS-induced mRNA expression of proinflammatory mediators in RAW264.7 cells.** RAW264.7 cells were pretreated with BBR (5 μM) for 1 h, then treated with palmitic acid (PA, 0.25 mM) or LPS (25 ng/mL) or both for 6 h. The mRNA levels of TNF-α, IL-6, MCP-1, and IL-1β were detected by real-time RT-PCR and normalized to HPRT1 as an internal control as described under Materials and Methods. Values are mean ± S.E. of three independent experiments. Statistical significance relative to vehicle control, **p<0.01, ***p<0.001; relative to LPS, $p<0.05; relative to PA+LPS, #p<0.05. **A**. TNF-α; **B**. IL-6; **C**. IL-1β; **D**. MCP-1.

IL-1β, and MCP-1 using real-time RT-PCR. As shown in Fig 1A, LPS significantly induced the expression of TNF-α. PA alone had minimal effect, but PA further promoted LPS-induced TNF-α expression, which was inhibited by BBR. Similarly, LPS/PA-induced upregulation of IL-6, IL-1β, and MCP-1 mRNA levels was also inhibited by BBR in mouse RAW264.7 macrophages (Fig 1B–1D).

In a parallel experiment, we measured the protein levels of TNF-α, IL-6, and MCP-1 secreted into cell culture media using ELISA. As shown in Fig 2A–2C, the protein levels of TNF-α, IL-6, and MCP-1 were significantly increased in the culture media of LPS-stimulated mouse RAW264.7 macrophages. PA alone also had no significant effect on protein expression of proinflammatory cytokines, but further potentiated LPS' effect. Both LPS and LPS+PA-induced upregulation of TNF-α, IL-6, and MCP-1 were significantly inhibited by BBR. In addition, we measured the mature IL-1β protein levels by Western blot. As shown in Fig 2D and 2E, the combination of LPS with PA-induced activation of the inflammasome and mature IL-1β procession, which was inhibited by BBR.

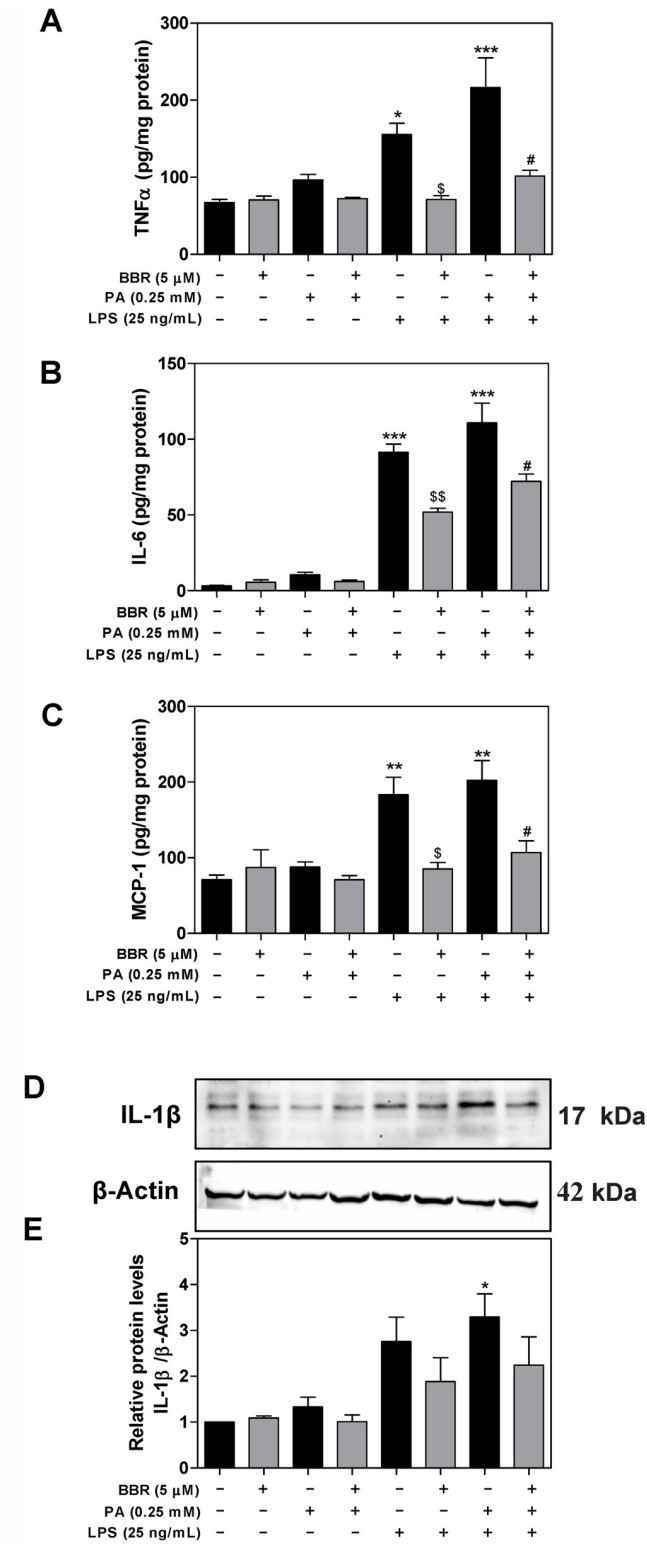

**Fig 2. Effect of BBR on PA and LPS-induced protein expression of proinflammatory mediators in RAW264.7 cells.** RAW264.7 cells were pretreated with BBR (5 μM) for 1 h, then treated with PA (0.25mM) or LPS (25 ng/mL) or both for 6 h. At the end of treatment, cell culture medium and total cell lysates were collected. The protein levels of TNF-α, IL-6, and MCP-1 were determined by ELISA as described under Materials and Methods. Relative protein levels of TNF-α, IL-6, and MCP-1 were normalized by total protein amounts and expressed as pg/mg of protein. The

protein levels of mature IL-1β were determined by Western blot. β-Actin was used as the loading control. Values are mean ±S.E. of three independent experiments. Statistical significance relative to vehicle control, *p<0.05, **p<0.01, ***p<0.001; relative to LPS, $p<0.05, $p<0.01; relative to PA+LPS, #p<0.05, ##p<0.01. **A**. The relative protein levels of TNF-α; **B**. The relative protein levels of IL-6; **C**. The relative protein levels of MCP-1; **D**. Representative immunoblots of mature IL-1β and β-Actin; **E**. The relative protein levels of IL-1β.

## 3.2. Effect of BBR on PA and LPS-induced UPR activation in RAW264.7 macrophages

It has become increasingly evident that sustained activation of the UPR signaling pathways and upregulation of transcription factors, such as CHOP and ATF4, contribute to ER stress-induced liver injury [21]. CHOP is the major mediator responsible for ER stress-induced apoptosis. We have previously reported that activation of ER stress contributed to the HIV protease inhibitor (ER stress-inducer)-induced inflammatory response [22]. Here, we further examined whether BBR has an inhibitory effect on PA and LPS-induced UPR activation in macrophages. As shown in Fig 3A, LPS and PA synergistically induced the expression of CHOP mRNA, which was markedly inhibited by BBR in RAW264.7 macrophages. LPS and PA also upregulated mRNA levels of ATF4 and XBP-1, but to less extent compared to CHOP. BBR also inhibited LPS and PA-induced ATF4 mRNA expression, but not XBP1 (Fig 3B and 3C).

In order to determine whether inhibition of the mRNA expression of CHOP and ATF4 by BBR is correlated to the reduction of protein levels, we measured the protein levels of CHOP, ATF4, and XBP1 by Western blot analysis. As shown in Fig 4, BBR significantly inhibited PA and LPS-induced protein expression of CHOP, ATF4.

## 3.3. Effect of BBR on PA and LPS-induced ERK activation in RAW264.7 macrophages

Activation of ERK1/2 has been reported to promote LPS-induced production of TNF-α, IL-6, IL-1β, and MCP-1 [23]. Our previous studies showed that BBR inhibits HIV protease inhibitor-induced inflammatory response by modulating ER stress signaling pathways in macrophages [12]. It also has been shown that BBR significantly inhibits the expression of inflammatory cytokines in ARPE-19 cells and that the inhibitory effect is mediated by inactivation of the ERK1/2, JNK, and p38 pathways [24]. In order to delineate the potential signaling pathways underlying the inhibitory effect of BBR on PA and LPS-induced TNF-α, IL-6, IL-1β, and MCP-1 expression and ER stress in macrophages, we further examined effects of PA/LPS and BBR on ERK activation. As shown in Fig 5, PA and LPS synergistically induced ERK activation, which was completely inhibited by BBR. As shown in the Supplementary S1 Fig, the effect of BBR on PA/LPS-induced ERK activation was time-dependent, the maximal effect was found at the 6 h time point. Furthermore, the inhibitory effect of BBR on PA/LPS-induced ERK activation was also dose-dependent (Fig 6).

## 3.4. Effect of BBR on PA and LPS-induced activation of inflammation, UPR, and ERK in primary mouse hepatocytes

To further examine the effect of BBR on PA/LPS-induced activation of inflammation, UPR, and ERK in hepatocytes, we isolated primary mouse hepatocytes and pre-treated with BBR for 1 h, then treated with PA/LPS for 6 h. The protein levels of TNF-a, IL-6, and MCP-1 were measured by ELISA. The protein levels of mature IL-1β, CHOP, ATF4, XBP1, p-ERK, and total

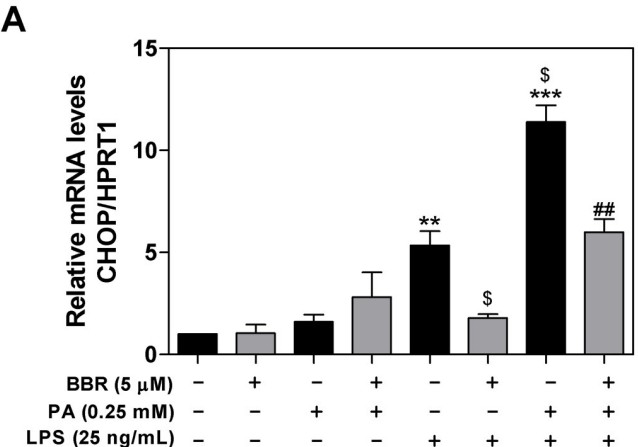

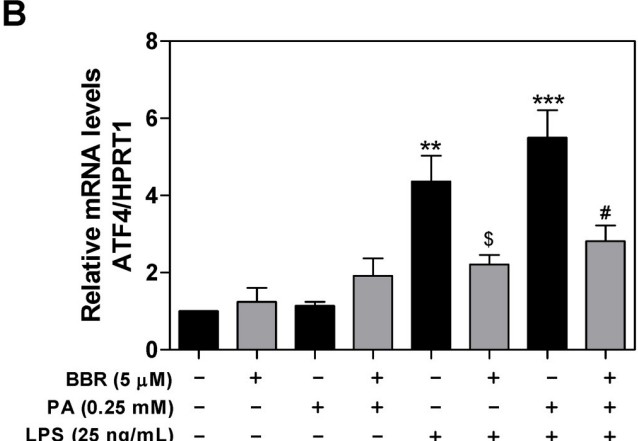

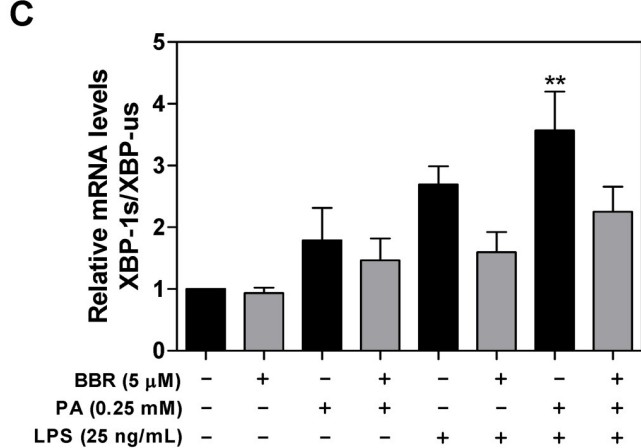

**Fig 3. Effect of BBR on PA and LPS-induced mRNA expression of the UPR genes in RAW264.7 macrophages.**
RAW264.7 cells pre-treated with BBR (5 μM) for 1 h, then treated with PA (0.25 mM) or LPS (25 ng/mL) or both for 6 h. The mRNA levels of CHOP, ATF4, XBP-1s, and XBP-1us were detected by real-time RT-PCR and normalized to HPRT1 as described under Materials and Methods. Values are mean ± S.E. of three independent experiments. Statistical significance relative to vehicle control, **p<0.01, ***p<0.001; relative to LPS, $p<0.05; relative to PA+LPS, #p<0.05, ##p<0.01. **A**. CHOP; **B**. ATF4; **C**. XBP-1s.

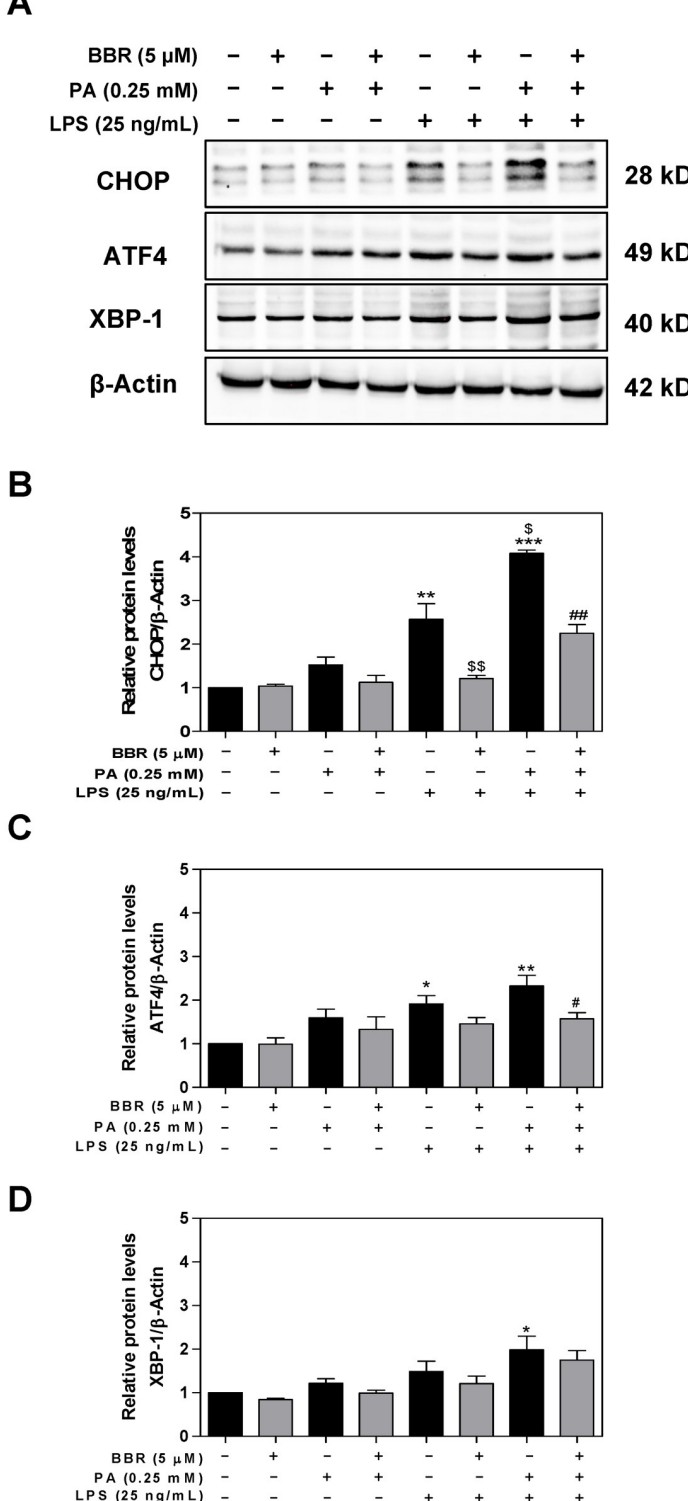

**Fig 4. Effect of BBR on PA and LPS-induced protein expression of the UPR genes in RAW264.7 macrophages.**
RAW264.7 cells were pre-treated with BBR (5 μM) for 1 h, then treated with PA (0.25mM) or LPS (25 ng/mL) or both for 6 h. Total cell lysates were prepared for Western blot analysis as described under Materials and Methods. β-Actin was used as the loading control. Values are mean ± S.E. of three independent experiments. Statistical significance relative to control, *p<0.05, **p<0.01; relative to PA+LPS, #p<0.05. **A**. Representative immunoblots of CHOP, ATF4, XBP-1s, and β-Actin; **B**. The relative protein levels of CHOP; **C**. The relative protein levels of ATF4; **D**. The relative protein levels of XBP-1s;.

**A**

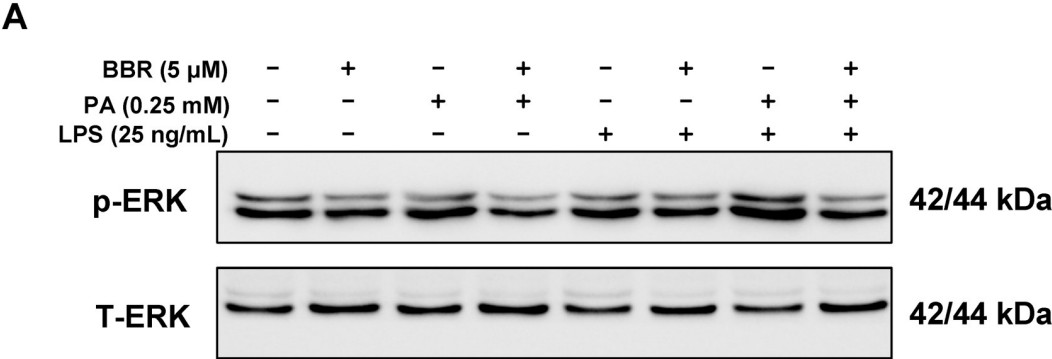

**B**

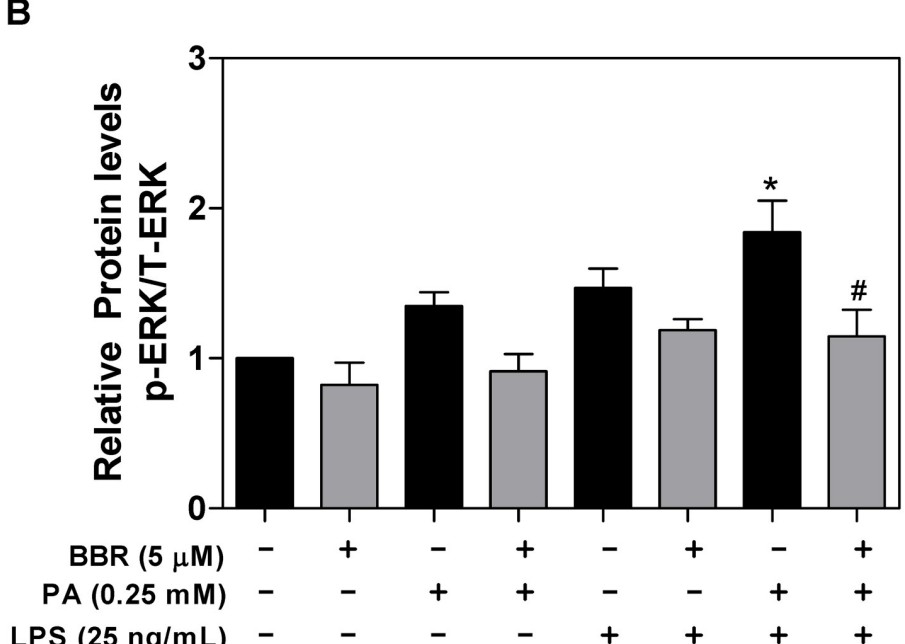

**Fig 5. Effect of BBR on PA and LPS-induced ERK activation in RAW264.7 macrophages.** RAW264.7 cells pre-treated with BBR (5 μM) for 1 h, then treated with PA (0.25mM) or LPS (25 ng/mL) or both for 6 h. Total cell lysates were prepared for Western blot analysis as described under Materials and Methods. Values are mean ± S.E. of three independent experiments. Statistical significance relative to vehicle control, *p<0.05, **p<0.01; relative to PA+LPS, #p<0.05. **A**. Representative immunoblots of phospho(p)-ERK and total (T)-ERK; **B**. The relative protein levels of p-ERK.

ERK were determined by Western blot analysis. As shown in Fig 7A–7C, PA/LPS-induced upregulation of the protein expression levels of TNF-α, IL-6, IL-1β, and MCP-1 were completely inhibited by BBR. The Western blot results further indicated that BBR not only inhibited PA/LPS-induced CHOP and ATF-4 activation but also significantly suppressed PA/LPS-induced ERK activation in primary mouse hepatocytes (Fig 7D–7G). However, LPS/PA had no effect on the protein expression levels of IRE1α, ATF6, and GRP78 (S2 Fig). Furthermore, the oil-red O staining showed that PA/LPS-induced hepatic lipid accumulation was also

**A**

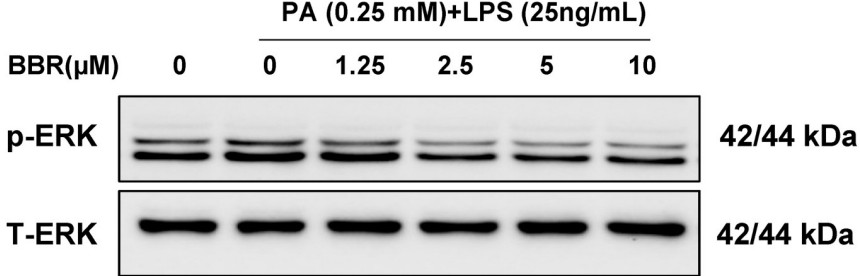

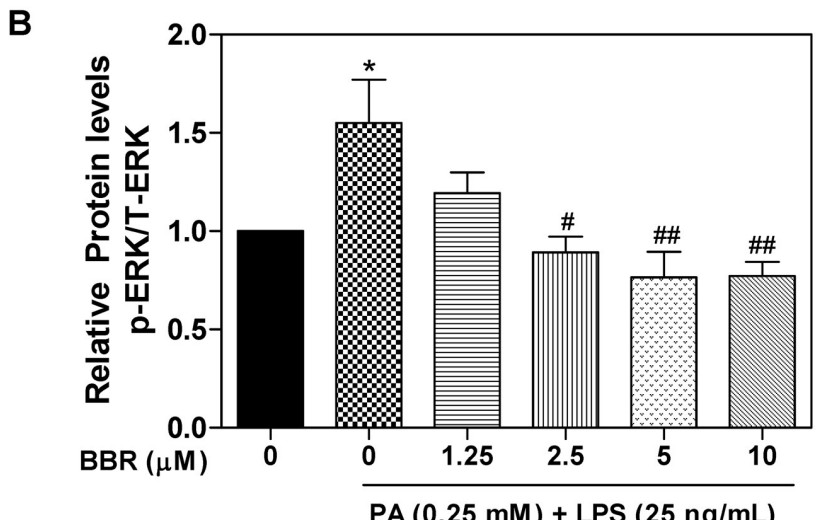

**Fig 6. Effect of BBR on PA and LPS-induced ERK activation in RAW264.7 macrophages.** RAW264.7 cells are pre-treated with BBR (0, 1.25, 2.5, 5, 10 μM) for 1 h, then treated with PA(0.25mM) and LPS (25 ng/mL) for 6 h. Total cell lysates were prepared for Western blot analysis as described under Materials and Methods. Values are mean ±S.E. of three independent experiments. Statistical significance relative to vehicle control, *p<0.05; relative to PA+LPS, #p<0.05, ##p<0.01. **A**. Representative immunoblots of phospho(p)-ERK and total (T)-ERK; **B**. The relative protein levels of p-ERK/total ERK.

inhibited by BBR (S3 Fig). However, LPS alone did not induce lipid accumulation, which was consistent with the previous report [25].

## Discussion

Macrophages play a critical role in activating the immune response against dangerous invaders, such as bacteria and viruses, by producing numerous proinflammatory mediators [26]. However, over activation of macrophages also causes tissue injury and promotes chronic disease progression, including metabolic liver disease, NAFLD. Therefore, inhibition of chronic inflammation becomes a potentially effective therapy to prevent the pathological progression of chronic diseases. In the rapid progression of the pandemic of obesity, NAFLD has emerged

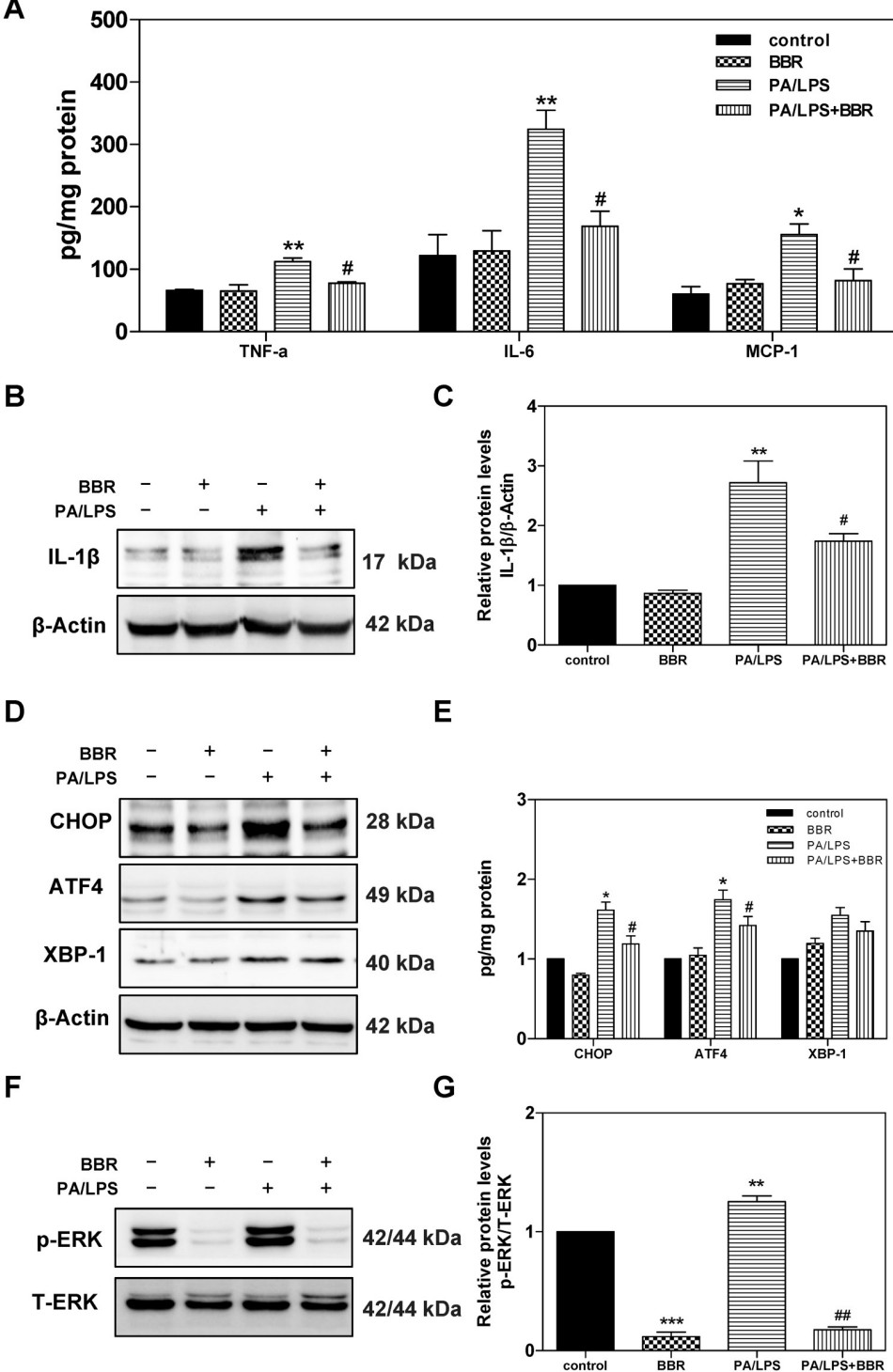

**Fig 7. Effect of BBR on PA and LPS-induced activation of inflammation, UPR, and ERK in primary mouse hepatocytes.** Primary mouse hepatocytes were isolated from C57BL/6 wild type mice and pre-treated with BBR (5 μM) for 1 h, then treated with PA (0.25mM) and LPS (25 ng/mL) for 6 h. At the end of treatment, cell culture medium and total cellular protein were collected. The protein levels of TNF-α, IL-6, and MCP-1 were determined by ELISA and were normalized by total protein amounts and expressed as pg/mg of protein. Total cell lysates were prepared for western blot analysis for IL-1β, CHOP, ATF4, XBP-1, phospho(p)-ERK, total (T)-ERK, and β-Actin. Values are mean ± S.E. of three independent experiments. Statistical significance relative to vehicle control, *p<0.05, **p<0.01; relative to PA+LPS, #p<0.05. **A**. The relative protein levels of TNF-α, IL-6, and MCP-1; **B**. Representative

immunoblots of IL-1β and β-Actin; **C**. The relative protein levels of IL-1β; **D**. Representative immunoblots of CHOP, ATF4, XBP-1s, and β-Actin. **E**. The relative protein levels of CHOP, ATF4, and XBP-1s; **F**. Representative immunoblots of phospho(p)-ERK and total (T)-ERK. **G**. The relative protein levels of p-ERK/total ERK.

as the most common chronic liver disease. Inflammation is a major contributor to insulin resistance, dyslipidemia, and metabolic syndrome [27,28]. BBR, isolated from *Rhizoma coptidis*, has been widely used in traditional Chinese medicine to treat bacteria infection for thousands of years. The clinically beneficial effects of BBR on metabolic diseases are linked to its anti-inflammatory activity [29–31].

Recent advances in NAFLD studies indicate that saturated fatty acid (SFA) or LPS promotes hepatic lipid accumulation and inflammation, which contribute to NAFLD disease progression [32–34]. Consistent with recent studies, our results showed that PA, a major SFA, synergistically promoted LPS-induced inflammatory response in both RAW264.7 macrophages and hepatocytes [35–37]. We have previously reported that the effect of BBR on high-fat diet-induced NAFLD was mediated by modulating gut microbiomes [38]. Numerous pre-clinical studies also showed a promising therapeutic effect of BBR on NAFLD by its modulation of inflammatory responses [39–41]. Consistent with previous findings, this study clearly indicated that PA exacerbated LPS-induced inflammation by increasing the mRNA and protein expression of proinflammatory mediators in macrophages and primary mouse hepatocytes, which was inhibited by BBR. Interestingly, PA alone did not significantly induce an inflammatory response in both macrophages and hepatocytes, but markedly promoted LPS-mediated activation of the inflammatory response. BBR efficiently blocked PA/LPS-induced upregulation of TNF-α, IL-6, IL-1β, and MCP-1 (Figs 1, 2 and 7).

As the major site in the cell for protein folding and trafficking, ER stress response has emerged as an essential cellular mechanism underlying numerous pathological conditions, such as inflammation and metabolic disorders [21,42,43]. Disruption of ER homeostasis leads to the activation of UPR. Extensive studies have shown that persistent activation of the UPR eventually induces inflammation and cell injury [6]. Our previous study reported that HIV protease inhibitors were strong ER stress inducers and activation of ER stress was responsible for HIV protease inhibitor-induced inflammation and dysregulation of lipid metabolism in macrophages and hepatocytes [12,44,45]. We also showed that BBR inhibited HIV protease inhibitor-induced inflammatory response by modulating ER stress response and inhibiting ERK activation in macrophages [12,46]. Here, we provide new evidence indicating that BBR-mediated beneficial effect against PA/LPS-induced inflammatory response is *via* modulating ER stress signaling pathways in both macrophages and hepatocytes. So far, three major branches of the UPR have been identified, including the IRE1 pathway, protein kinase RNA-like ER kinase (PERK) pathway, and ATF6 pathway [21]. Our results in the current study demonstrated that BBR significantly inhibited PA/LPS-induced activation of the PERK-ATF4-CHOP signaling pathway in macrophages and primary mouse hepatocytes, but no significant impact on protein levels of ATF6, IRE1α, and GRP78 in RAW264.7 macrophages (S2 Fig). Our previous study showed that activation of CHOP is responsible for ERK activation and subsequent upregulation of the expression levels of proinflammatory mediators in macrophages [46]. We also showed that knockout CHOP reduced ER stress-induced hepatic dyslipidemia and intestinal barrier dysfunction [47,48]. Most recent studies with high fat diet-induced NAFLD rodent models and Larval Zebrafish model indicated that several signaling pathways, such as the nuclear factor erythroid 2-related factor 2/antioxidant response element (Nrf2/ARE), sirtuin 3 (SIRT3)/AMPK/ACC, and AMPK-SREBP1c-SCD-1 pathways, are potential targets [39,41,49–51].

In summary, the current study identified a key cellular mechanism underlying the potential protective effect of BBR on PA and LPS-induced inflammatory response in macrophages and hepatocytes. BBR is an effective ER stress modulator, and its beneficial effects on preventing inflammatory and metabolic diseases may be largely through regulating the UPR signaling pathways. Therefore, further *in vivo* study using a clinically relevant NAFLD/NASH model and clinical studies are needed to evaluate the potential applications of BBR as a therapeutic agent for NAFLD.

## Supporting information

**S1 Fig. Effect of BBR on PA and LPS-induced ERK activation in RAW264.7 macrophages.** RAW264.7 cells were pre-treated with BBR (5 μM) for 1 h, then treated with PA(0.25mM) and LPS (25 ng/mL) for 2, 6, 12 and 24h. Total cell lysates were prepared for Western blot analysis as described under Materials and Methods. Values are mean ± S.E. of three independent experiments. Statistical significance relative to vehicle control, $^*p<0.05$; relative to PA+LPS, #p$<0.05$, ##p$<0.01$. A. Representative immunoblots of phospho(p)-ERK and total (T)-ERK; B. The relative protein levels of p-ERK.
(PDF)

**S2 Fig. Effect of BBR on PA and LPS-induced protein expression of IRE1α, ATF6, and GRP78 in RAW264.7 macrophages.** RAW264.7 cells are pre-treated with BBR (5 μM) for 1 h, then treated with PA (0.25 mM) and LPS (25 ng/mL) for 6 h. Total cell lysates were prepared. The protein expression levels of IRE1α, ATF6, GRP78, and β-Actin were measured by Western blot analysis as described under Materials and Methods. β-Actin was used as the loading control. **A**. Representative immunoblots of IRE1α, ATF6, GRP78, and β-Actin are shown.
(PDF)

**S3 Fig. Effect of BBR on PA and LPS-induced lipid accumulation in mouse primary hepatocytes.** Primary mouse hepatocytes were plated on 22 × 22-mm glass coverslips in 6-well plates. Hepatocytes were pre-treated with BBR (5 μM) for 1 h, then treated with PA (0.25 mM) or LPS (25 ng/mL) or both for 6 h. At the end of the treatment, hepatocytes were fixed with 3.7% formaldehyde in PBS for 30 min followed by two washes with PBS. The hepatocytes were stained with 0.2% Oil Red O in 60% 2-propanol for 10 min and washed three times with PBS. The images of Oil Red O staining were taken with a microscope (Olympus, Tokyo, Japan) equipped with an image recorder under a 10 × lens. **A**. DMSO; **B**. BBR; **C**. PA; **D**.PA+BBR; **E**. LPS; **F**.LPS+BBR; **G**.PA/LPS; **H**. PA/LPS+BBR.
(PDF)

**S1 Table. List of antibodies.**
(DOCX)

**S2 Table. List of QPCR primers.**
(DOCX)

**S1 Raw image.**
(PDF)

## Author Contributions

**Conceptualization:** Yanyan Wang, Xiqiao Zhou, Weidong Chen, Huiping Zhou.

**Data curation:** Yanyan Wang, Derrick Zhao, Xuan Wang, Emily C. Gurley.

**Formal analysis:** Yanyan Wang.

**Funding acquisition:** Huiping Zhou.

**Methodology:** Yanyan Wang, Xiqiao Zhou, Derrick Zhao, Xuan Wang, Emily C. Gurley, Runping Liu, Xiaojiaoyang Li.

**Project administration:** Huiping Zhou.

**Supervision:** Phillip B. Hylemon.

**Writing – original draft:** Yanyan Wang.

**Writing – review & editing:** Phillip B. Hylemon, Weidong Chen, Huiping Zhou.

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
