## [Decision Letter · Decision Letter 0]

23 Mar 2020

PONE-D-20-07442

Berberine inhibits free fatty acid and LPS-induced inflammation via modulating ER stress response in macrophages and hepatocytes

PLOS ONE

Dear Dr. Zhou,

Thank you for submitting your manuscript to PLOS ONE. After careful consideration, we feel that it has merit but does not fully meet PLOS ONE’s publication criteria as it currently stands. Therefore, we invite you to submit a revised version of the manuscript that addresses the points raised during the review process.

We would appreciate receiving your revised manuscript within 30 days; congratulations on this wonderful study. To enhance the reproducibility of your results, we recommend that if applicable you deposit your laboratory protocols in protocols.io, where a protocol can be assigned its own identifier (DOI) such that it can be cited independently in the future. For instructions see: http://journals.plos.org/plosone/s/submission-guidelines#loc-laboratory-protocols

We look forward to receiving your revised manuscript.

Kind regards,

Gianfranco D. Alpini

Academic Editor

PLOS ONE

Journal Requirements:

2. To comply with PLOS ONE submission requirements, in your Methods section, please provide additional information regarding the experiments involving animals and ensure you have included details on (1) methods of sacrifice, (2) methods of anesthesia and/or analgesia, and (3) efforts to alleviate suffering.

Reviewers' comments:

Reviewer's Responses to Questions

**Comments to the Author**

1. Is the manuscript technically sound, and do the data support the conclusions?

Reviewer #1: Yes

2. Has the statistical analysis been performed appropriately and rigorously? 

Reviewer #1: Yes

3. Have the authors made all data underlying the findings in their manuscript fully available?

Reviewer #1: Yes

4. Is the manuscript presented in an intelligible fashion and written in standard English?

Reviewer #1: Yes

5. Review Comments to the Author

Reviewer #1: The current study showed that PA, a major SFA, synergistically promoted LPS-induced inflammatory response, ER stress in both RAW264.7 macrophages and primary mouse hepatocytes, and BBA, an herbal extract from Rhizoma coptidis, could repress the PA-LPS induced inflammation and induction of ER stress-related factors at both mRNA and protein levels. Liver fat accumulation and inflammation are two major factors in driving the progression from simple hepatic steatosis to steatohepatitis. There is no FDA approved medicine to prevent or treat non-alcoholic steatohepatitis (NASH) a more severe stage in the spectrum of non-alcoholic fatty liver diseases (NAFLD). In this regard, this study provide insight in using BBR as a preventative agent in preventing NAFLD transition to NASH in vivo.

This study is well designed and conducted. The writing is clear and easy to follow. The current study showed specific modulation of the ER stress pathways by BBR with suppressing ERK activation as the potential molecular mechanism.

A minor recommendation is to further provide the known mechanisms of action of BBR in the context of NAFLD and NASH studies, and how this study provides in vitro evidence supporting or contracting the known effects of BBR on liver lipid metabolism and inflammation.

6. PLOS authors have the option to publish the peer review history of their article (what does this mean?). If published, this will include your full peer review and any attached files.

Reviewer #1: No

---

## [Author Response · Author response to Decision Letter 0]

12 Apr 2020

Response to Reviewer’s comment

Comment: A minor recommendation is to further provide the known mechanisms of action of BBR in the context of NAFLD and NASH studies, and how this study provides in vitro evidence supporting or contracting the known effects of BBR on liver lipid metabolism and inflammation.

Response: We would like to thank the Reviewer for his or/her time spent reviewing our manuscript and helpful comment. In the revised manuscript, we provided the information regarding the potential mechanisms of Berberine-mediated therapeutic effect on NAFLD/NASH based on the most recent studies in the discussion.

---

## [Editor Report · Decision Letter 1]

20 Apr 2020

Berberine inhibits free fatty acid and LPS-induced inflammation via modulating ER stress response in macrophages and hepatocytes

PONE-D-20-07442R1

Dear Dr. Zhou

We are pleased to inform you that your manuscript has been judged scientifically suitable for publication and will be formally accepted for publication once it complies with all outstanding technical requirements.

With kind regards,

Gianfranco D. Alpini

Academic Editor

PLOS ONE
---

## [Editor Report · Acceptance letter]

22 Apr 2020

PONE-D-20-07442R1 

Berberine inhibits free fatty acid and LPS-induced inflammation *via* modulating ER stress response in macrophages and hepatocytes 

Dear Dr. Zhou:

I am pleased to inform you that your manuscript has been deemed suitable for publication in PLOS ONE. Congratulations! Your manuscript is now with our production department. 

With kind regards,

on behalf of

Dr. Gianfranco D. Alpini 

Academic Editor

PLOS ONE